# The Linea Alba Width, Children’s Physical Activity, and Chosen Anthropometric Measurements: The Results of the Cross-Sectional Study

**DOI:** 10.3390/pediatric17010025

**Published:** 2025-02-18

**Authors:** Agata Maria Kawalec-Rutkowska, Agata Marczak, Marian Simka

**Affiliations:** Department of Anatomy, Institute of Medical Sciences, University of Opole, 45-052 Opole, Poland; amarczak@uni.opole.pl (A.M.); msimka@uni.opole.pl (M.S.)

**Keywords:** child, diastasis recti, physical activity, body composition

## Abstract

**Objectives:** This study was aimed at the assessment of the relationship between the presence of diastasis recti abdominis in children and the clinical variables potentially attributable to the wider linea alba. **Methods:** Fifty-one children, aged 8–12 years, were evaluated. The study protocol included ultrasonographic measurements of the linea alba width, anthropometric measurements, body composition assessment with the use of the Tanita Body Composition Analyzer, and the questionnaire assessing clinical history and the level of physical activity. **Results:** Statistical analysis revealed that the interrectus distance, which was ≥20 mm, was significantly more often found in boys, in children with a higher body length at birth and a higher waist/hip ratio, and also in those with a history of congenital umbilical hernia. Other variables, such as the level of physical activity, body weight, parameters of the body composition measured with the body analyzer, presence of abdominal symptoms (abdominal pain, constipation, urinary incontinence), or family history of musculoskeletal disease, were not associated with the presence of diastasis recti abdominis. **Conclusions:** The results of our study suggest a congenital background of diastasis recti abdominis in children, especially given its higher prevalence among boys and those children who presented with specific body parameters at birth.

## 1. Introduction

The linea alba is a fibrous structure that separates two rectus abdominis muscles and extends from the xiphoid process of the sternum to the superior pubic ligament. Three different zones, characterized by fiber orientation, can be distinguished within the linea alba: the lamina fibrae obliquae, the lamina fibrae transversae, and a small lamina fibrae irregularium [1]. The oblique fibers play a role in trunk movements, while the transverse fibers are a counterpart to the intraabdominal pressure [1]. The diastasis recti abdominis (DRA) is diagnosed when there is an abnormally wide distance between the rectus abdominis bellies, yet without associated fascia defect [2].

It is described as a separation of the rectus abdominis muscles along the linea alba that can result in a midline abdominal bulge [3]. It must be underlined that this condition is not a hernia, and there is no risk of incarceration [4]. It is crucial to distinguish whether the fascia is intact or disrupted because this is the difference between DRA and a hernia of the linea alba [3].

Diagnostic techniques for diagnosing DRA include physical examination and imaging studies [3].

The recommended way is to examine patients in the supine position and ask them to contract their abdominal muscles (during the half sit-up position or with a leg raised) [3]. In a majority of normally-weighted patients, such a physical examination is sufficient to confirm the diagnosis of DRA [3,4,5,6]. It can be more difficult in obese individuals [3]. Digital calipers and diastomameters have also been reported as diagnostic methods [7].

The imaging modalities used in the diagnostic workout include ultrasound, computed tomography (CT), and magnetic resonance imaging (MRI) [3]. Ultrasound is considered the cheapest and the most accessible imaging modality [3]. When considering the measurement in children, it is worth emphasizing that this method neither requires anesthesia nor exposure to ionizing radiation. According to Petronila et al., palpation techniques, digital calipers, and diastometers are reliable tools but cannot replace ultrasound imaging during clinical measurement of DRA [7].

Imaging methods determine the severity of DRA by measuring the interrectus distance (IRD) [3,4,5,6]. The IRD is defined as the width of the linea alba between the aponeurotic sheaths surrounding the paired rectus abdominis muscles [8].

To visualize the rectus abdominis separation in the ultrasound, a high-frequency linear array probe should be used; using the muscular preset is recommended [6]. During measurement, the patient should not contract these muscles [6]. This means lying in the supine position with the abdominal muscles fully relaxed [6]. The probe plane should be perpendicular to the long axis of the abdomen [6]. In most studies, the IRD measurements were performed at the level of the umbilicus and 2 or 3 cm above and below [6,9].

It should be underlined that there are no generally accepted criteria for the diagnosis of DRA in adults and children [4]. The need for DRA screening method standardization and IRD measurement protocol standardization are widely discussed [9]. Opala et al. conducted a scoping review according to the PRISMA-ScR guidelines and noticed discrepancies between the IRD measurement procedures, which made comparing the studies difficult [9].

There are many classifications used to assess the severity of DRA in adults, for example, the Rath classification and the Nahas classification [3]. Thus far, there are no classifications for children. In adult patients, IRD, which is ≥ 2 cm wide, is considered clinically significant [3]. In the literature, the cutoff points of the IRD values in children, such as 15 mm and 20 mm, were considered [10]. The choice of 20 mm as the cutoff point seems more appropriate because Cohen’s kappa coefficient for the diagnosis of DR using the clinical definition and ultrasonographic criterion (20 mm) was higher than for the ultrasonographic criterion (15 mm), indicating a higher agreement between these two modes of diagnosing [10]. Risk factors for DRA in adults comprise conditions weakening the linea alba, such as multiple pregnancies, obesity, or previous abdominal surgery [11,12]. However, epidemiological data on the prevalence of DRA in the general population are quite scarce. Such studies were limited to adult patients, particularly pregnant and postpartum women, obese individuals, and cadavers [12]. Of note, a generally accepted cutoff of 20 mm used to define DRA is probably too small. Kaufmann et al. noticed that the prevalence of DRA in adults, defined by the above-mentioned value, can be as high as 57% and suggested a revision of the definition of DRA to avoid over-treatment [12]. Moreover, Cavalli et al. claimed that DRA should rather be considered an anatomic variant and not a pathology by itself [2].

Although DRA is considered to be an acquired condition [1], it can also be found in children. In comparison with the adults, data on DRA in the pediatric population are even more limited. It is known that DRA is more prevalent in pediatric patients presenting with genetic disorders. Still, an enlarged distance between two rectus abdominis muscles can also be found among otherwise healthy children.

DRA seems to be a quite common finding in children, especially among boys [10]. There is little information in the literature regarding the clinical significance of the presence of DRA in children. It has been described that only a few healthy children with DRA reported abdominal pain after physical exertion, constipation, and fecal and urinary incontinence, but statistically sound conclusions could not be drawn [10]. It was demonstrated that the width of the linea alba in children can be related to the spine’s shape [13]. Zmyślna et al. found that children with DRA present with a posterior shift of the body axis [13]. Consequently, DRA can be considered a sign of poor coordination of the abdominal muscles, which are important stabilizers of the spine [13]. In current pediatric research on DRA, there are no studies comparing the body composition of children with and without DRA and their physical activity. This knowledge gap may lead to a lack of consistent consensus among pediatricians and physiotherapists, confusion among caregivers, as well as to the overtreatment and unnecessary rehabilitation of healthy children. This study was aimed at assessing the relationship between the presence of DRA in children and other clinical parameters, such as anthropometric measurements, physical activity levels, family history of musculoskeletal diseases, and other clinical variables potentially attributable to DRA.

## 2. Materials and Methods

Participants of this study were recruited in schools and through advertisements on social media. Sixty-five children aged 8–12 years met the inclusion criteria of this study. Questionnaires, anthropometric measurements, and ultrasonographic examinations were conducted from July 2024 to September 2024 in the Department of Anatomy of the University of Opole. This study was performed in line with the Declaration of Helsinki. The study protocol has been approved by the Bioethical Committee of the University of Opole (approval No. UO/0004/KB/2024). Informed consent was obtained from all parents or legal guardians of the children participating in this survey.

The inclusion criteria of the study comprised the following:Children aged 8–12 years;Maturity of a child, allowing participation in the study, especially regarding ultrasonographic examination and bioelectrical impedance analysis;Informed consent given by a parent or legal guardian.

The exclusion criteria comprised the following:History of abdominal surgery;History of cardiosurgical procedures, especially of implantation of cardiac electronic devices, such as pacemakers or implantable defibrillators;Previous or active gastrointestinal disease;History of neurological disease (e.g., cerebral palsy or muscular dystrophy);Abdominal hernia (active or after treatment) and other pathologies associated with weakness of the abdominal muscles.

The recruitment process included sending the information about the study to the secretariats of 23 public primary schools in Opole and disseminating the information on a Facebook page dedicated to mothers of children from Opole (“Mamy z Opola group”). The caregivers interested in participating in this study used the study’s website to answer questions about the exclusion criteria and to choose a date and hour of the examination. To avoid bias, clear inclusion and exclusion criteria were applied before recruitment, the information was sent to all the primary schools in the region, and the study’s purpose and procedures were explained to the participants and their guardians to encourage informed and voluntary participation. The questionnaire, answered by the child’s parent or legal guardian, included questions regarding gestational age at birth, body mass at birth, body length at birth, concomitant diseases, previous surgical treatment, and medications. Also, there were questions about symptoms such as abdominal pain during or just after physical exertion, constipation, fecal and/or urinary incontinence, and family history of muscle disease. In addition, the parent and the child answered the questions from the PAQ-C (The Physical Activity Questionnaire for Older Children) in the presence of the researcher conducting the study [14,15,16,17].

The anthropometric measurements included four parts.

The height was measured with a stadiometer SECA-213 (SECA, Hamburg, Germany) with an accuracy of 1 mm. Before taking the measurements, the child was asked to remove any footwear and/or head ornaments [18,19,20]. The measurements were taken from the buttocks, blades of the scapulae, and the back of the head located next to the stadiometer board; the head was oriented in the Frankfurt horizontal plane (FH plane; the horizontal line from the ear canal to the lower border orbit was parallel to the floor; the line also intersected the vertical backboard perpendicularly) [18,19,20]. During the measurements, the child was asked to take a deep breath and to stand as straight as possible (deep breath straightens the spine and allows for more accurate and consistent measurements) [18,19,20]. The stadiometer headpiece was firmly placed on the head, and the results of the measurements were noted to the nearest tenth of a centimeter (1 mm).The body mass and body composition values were measured with the Tanita Body Composition Analyzer DC-430MA (Tanita Europe, Amsterdam, the Netherlands) with an accuracy of 0.1 kg. According to the information given by the producer, this analyzer can be used in children older than 5 years [21]. During the measurements, the child was standing without bending the knees and with feet parallel to the electrodes. Before the examination, the parents were instructed that the child should not eat or drink too much the day before the measurement, should not eat or drink 3 h before the assessment, and should urinate just before the visit. In the case of girls, visits were not scheduled during menstruation [21]. With the use of the above-described analyzer (using bioelectrical impedance analysis—BIA), the following measurements were performed: body weight [kg], body fat mass (fat mass), body fat % (fat%), body muscle mass (muscle mass), total body water (TBW), total body water % (tbw%), and fat-free mass (FFM).The waist, abdominal, and hip circumferences were measured with the measuring tape with an accuracy of 1 mm. The waist circumference was measured at the minimum circumference between the iliac crest and the rib cage at the end of normal exhalation, with an accuracy of 0.1 cm (the same definition was used in the OLAF study, which was aimed at establishing the normal values for Polish children) [18,19,20]. The abdominal circumference was measured at the level of the umbilicus, with an accuracy of 1 mm (the definition used by the WHO). The hip circumference measurements were taken around the widest portion of the buttocks (the definition used by the WHO and in the OLAF study) [18,19,20].Ultrasonographic examinations were conducted by the same physician. It was performed with the use of the GE Versana Active set (GE HealthCare, Chicago, IL, USA), with a 10 MHz linear probe and using the muscular preset. The interrectus distance (IRD)—the distance between two bellies of the rectus abdominis muscles, was measured at 5 points:Xiphoid (X)—defined as the point just below the xiphoid process;Xipho-umbilical (XU)—defined as the point 2 cm above the umbilicus;Umbilical (U)—defined as the point at the level of the umbilicus;Pubo-umbilical (PU)—defined as the point 2 cm below the umbilicus;Pubical (P)—defined as the point just above the pubic symphysis.

To maintain consistency, the same protocol and techniques were followed during each examination. Firstly, the patients were examined on the same examination table in the supine position; the measurement sites were marked with the body marker. During measurement, the probe plane was perpendicular to the long axis of the abdomen, and image acquisition was performed at the end of normal expiration. All these ultrasonographic measurements were taken three times, and the mean values were used in further analysis.

Secondly, the examination was performed in the sitting and standing positions on the site, which was marked previously. All ultrasonographic examinations were performed by physicians with experience in ultrasonography, including certificates in pediatric ultrasound examinations and certificates of expertise given by the Polish Ultrasound Society. After the pseudo-anonymization of the anthropometric data, the body mass index (BMI) and waist-to-hip ratio (WHR) were calculated. For each individual studied, the percentiles of the body mass, height, and body mass index were established according to the OLAF calculator for the Polish population. The values of BMI were then categorized into the following groups according to the OLAF calculator for the Polish population: underweight (below the 5 percentile), normal (5–85 percentile), overweight (≥85 percentile), and obese (≥95 percentile) [18,19,20]. We also assessed the level of physical activity of each child using the PAQ-C (The Physical Activity Questionnaire for Older Children). The items of this questionnaire were scored on a 5-point scale. Then, the final PAQ-C activity summary score was calculated. A score of 1 indicated low physical activity, whereas a score of 5 indicated high physical activity [14,17].

The Physical Activity Questionnaire for Older Children was already validated for the Polish population and showed a high reliability and internal consistency in measuring general physical activity levels in youth [14,15,16,17,22,23,24,25]. Regarding the clarification of the handling of missing data, the parent was given a paper questionnaire to fill out in the presence of the researcher. The questionnaire was then checked for completeness. The parent was also asked to bring the child’s health booklet, which contains important information such as birth length, gestational age at the time of birth, birth weight, and key medical conditions. This approach helped minimize missing data. Additionally, the PAQ-C questionnaire was completed in the presence of the researcher, in line with the recommendations, and was reviewed by the researcher after the parent had filled it out.

For the purpose of this survey, the enlarged IRD was defined as wider than 20 mm at any of the above-described body positions and parts of the linea alba.

The choice of this cutoff point was driven by the results of our previous study, which demonstrated that Cohen’s kappa coefficient for the diagnosis of DR using the clinical definition and ultrasonographic criterion (20 mm) was higher than for the ultrasonographic criterion (15 mm) (0.49 vs. 0.32), indicating higher agreement between these two modes of diagnosing [10]. A higher kappa suggests that the assessment process is more reliable and consistent, reducing the potential for errors or discrepancies in the data.

Statistical analysis was performed with the use of Excel and the PAST data analysis package (version 3.0; University of Oslo, Norway). The normal distribution of parametric variables was assessed with the use of the Shapiro–Wilk test. The two-sample Mann–Whitney test was used to assess the potential association between the IRD and quantitative variables when the distribution was non-normal. The t-test was used for quantitative variables exhibiting normal distribution. To assess the potential associations between the IRD and categorical variables, such as gender or preterm birth, the chi-square/Fisher’s exact test was used. The significance of all statistical tests was set at *p* < 0.05. The power calculations were performed using G*Power Version 3.1.9.7 (Franz Faul, Christian-Albrechts-Universität, Kiel, Germany); Power (1-β) = 0.79 ≈ 0.8. However, it should be underlined that in order to optimize the reliability of statistical analyses and establish the norms for IRD, it would be necessary to increase the sample size, which, with this method, has been calculated to be 385 children. The sample size in this study is too small to draw a valid conclusion on the cohort studied.

## 3. Results

### 3.1. The Study Group

Out of 65 participants willing to participate in the study, four children were excluded due to previous abdominal surgical treatment, one child due to neurologic disease (epilepsy), two due to the infectious diseases present on the day of measurements, and seven caregivers refused to give informed consent. Finally, 51 children (20 girls and 31 boys) were included and assessed. The anthropometric measurements and the PAQ-C values are presented in Table 1.

**Table 1 pediatrrep-17-00025-t001:** The anthropometric and the PAQ-C activity summary score (The Physical Activity Questionnaire for Older Children) parameters in the study group.

Parameter	Mean	Median	SD	Min	Max
PAQ-C activity summary score	3.08	3.02	0.56	2.05	4.34
Age (years)	9.27	9.00	1.10	8.00	12.00
Body mass (kg)	35.84	31.30	13.81	22.40	114.00
Body mass (centile)	50.27	49.00	29.13	0.10	99.90
Height (cm)	138.80	137.00	9.33	120.00	165.00
Height (centile)	47.63	43.00	28.99	1.00	97.00
Body Mass Index—BMI (kg/m^2^)	18.24	17.29	4.86	13.17	46.84
Body Mass Index—BMI (centile)	53.02	56.00	28.67	0.10	99.90
Abdominal circumference (cm)	64.53	63.00	7.66	52.00	83.00
Waist circumference (cm)	60.48	58.00	6.44	50.00	77.00
Hip circumference (cm)	69.16	68.00	7.49	57.00	88.00
Waist-to-Hip Ratio WHR	0.88	0.88	0.04	0.80	1.03
Body fat %—Fat%	19.15	17.90	6.31	8.90	36.70
Body fat mass (Fat mass)	7.24	6.00	4.24	2.50	22.90
Fat-free mass (FFM)	27.42	26.20	6.20	6.90	43.80
Muscle mass	26.28	24.80	5.22	18.70	41.50
Total body water (Tbw)	20.43	19.50	4.01	14.60	32.10
Total body water % (Tbw%)	58.50	60.10	7.37	17.10	66.50

The average values of the ultrasonographic measurements are given in Table 2. An example of an ultrasonographic picture is presented in Figure 1.

**Figure 1 pediatrrep-17-00025-f001:**
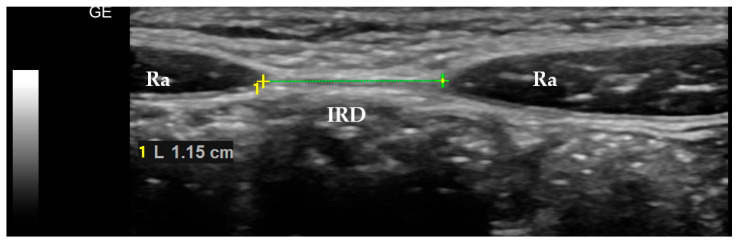
Ultrasonographic measurement of the linea alba. Ra—rectus abdomis muscle; IRD—interrectus distance.

### 3.2. The Anthropometric Parameters in Children and the Interrectus Distance

Statistical analysis of the anthropometric vs. clinical parameters revealed that the IRD, which was wider than 20 mm in any of the body positions, was significantly more often found in children who had a higher body length at birth (55.83 ± 2.52 vs. 53.37 ± 3.32; *p* = 0.023; the 95% confidence interval (CI 95%) for the difference between groups was [53.063–54.857]) and a higher waist/hip ratio (0.87 ± 0.03 vs. 0.90 ± 0.06; *p* = 0.009; CI 95% [86.902–89.098]). The details are given in Table 3.

In addition, the IRD, which was wider than 20 mm, was significantly more frequently found among boys (91.67% vs. 8.33%; *p* = 0.017; Figure 2).

On the other hand, there were no statistically significant differences regarding an enlarged vs. normal IRD between children presenting with particular categories of BMI. Among underweighted children, the prevalence of an enlarged vs. normal IRD was 16.17% vs. 5.13%; among normal-weighted children, 50.0% vs. 82.05%; among overweighted ones, 16.67% vs. 10.26%; and among obese children: 16.67% vs. 2.56%. These differences were insignificant (Fisher’s exact test *p*-value: 0.074).

### 3.3. Clinical Symptoms and Medical History in Children and the Interrectus Distance

Except for the history of umbilical hernia after birth, which was more often present among children with an enlarged IRD (33.33% vs. 5.13%; *p* = 0.022; Figure 3), other clinical variables were not associated with an enlarged IRD. These irrelevant symptoms and clinical history comprised abdominal pain during or just after physical exercise (*p* = 0.845), constipation (*p* = 0.673), urinary incontinence (*p* = 0.247), and family history of musculoskeletal disease (*p* > 0.05).

### 3.4. Physical Activity Levels in Children and the Interrectus Distance

Then, we assessed an association between physical activity and DRA. The level of physical activity, measured with the PAQ-C score, varied from 2.05 to 4.34. In this study cohort, the girls were less active than the boys (2.84 ± 0.53; median: 2.76; range: 2.05–3.72 vs. 3.23 ± 0.53; median: 3.06; range: 2.31–4.34), and this difference was statistically significant (*p* = 0.019; 95%CI: [2.926–3.234]). The details are shown in Figure 4 and Figure 5.

However, a normal and widened IRD were equally often found among children reporting low, moderate, or high levels of physical activity, assessed with the PAQ-C (Figure 6, Table 3; *p* = 0.886).

The comparison between the group of children with high and low activity did not reveal significant differences in terms of the prevalence of IRD, which exceeded 20 mm (*p* = 0.924).

The analysis of body composition by BIA did not reveal significant differences between the group with an IRD < 20 mm and ≥ 20 mm. On the other hand, children with high activity levels presented with a lower BMI, lower fat %, fat mass, and fat-free mass when compared to those less active. Still, these differences did not regard IRD (Table 4). It should be emphasized that according to the manual of the Tanita analyzer, the parameter fat-free mass consists of the muscle, bone, tissue, water, and all other fat-free mass in the body [11]. Moreover, according to this manual, the parameter muscle mass is described as bone-free lean tissue mass (LTM) [11]. Of note, in the pediatric cohort studied, children with different levels of physical activity exhibited a similar muscle mass % (*p* > 0.05).

### 3.5. The Analysis of Gender Differences

The analysis of gender differences in the children revealed that boys and girls from our study group did not differ regarding age, body mass, body mass centile, body mass index (BMI), BMI centile, height, height centile, abdominal circumference, and hips circumference (*p* > 0.05). Surprisingly, the WHR was higher in boys than in girls (0.89 ± 0.04 vs. 0.85 ± 0.03; *p* = 0.001).

The genders did not differ regarding the presence of abdominal symptoms such as abdominal pain, constipation, and urinary incontinence (*p* > 0.05).

The analysis of data regarding preterm birth, birth weight, and the presence of umbilical hernia after birth showed no significant differences between genders. The body length at birth was higher in boys than in girls (*p* = 0.04).

The analysis of the ultrasonographic examination results revealed that DRA (the IRD wider than 20 mm) was significantly more frequently found among boys (91.67% vs. 8.33%; *p* = 0.017; CI 95% [1.267–1.553]). Moreover, a detailed analysis revealed significant differences in the IRD in points XU and U in the supine, sitting, and standing positions (Table 5). No differences were found regarding measurements of the IRD in the X, PU, and P points.

The PAQ-C score was higher in boys than in girls (*p* = 0.019). This means the girls were less active than the boys (2.84 ± 0.53; median: 2.76; range: 2.05–3.72 vs. 3.23 ± 0.53; median: 3.06; range: 2.31–4.34).

The analysis of differences in the body composition revealed no differences in fat mass, FFM, or muscle mass (*p* > 0.05). However, the girls had a higher fat % (21.85 ± 6.98 vs. 17.41 ± 5.13; *p* = 0.011) and lower percentage of total body water (55.5 ± 10.1 vs. 60.44 ± 3.75, *p* = 0.026) than boys.

## 4. Discussion

In this study, we found that an increased IRD was more prevalent in boys, in children with a higher body length at birth, and in those presenting with a higher waist/hip ratio. Other parameters, such as the BMI, body composition measured with the Tanita Body Composition Analyzer, clinical and family histories (except for a history of congenital umbilical hernia), and the level of physical activity, were not correlated with the presence of DRA. Our study is one of very few performed in the pediatric population. Moreover, an assessment of a potential link between physical activity levels and the presence of DRA in children has not been investigated before.

In the literature, DRA is considered an acquired condition in which the rectus abdominis muscles are abnormally separated without fascia defect [2]. However, our results suggest that, at least in children, DRA should rather be regarded as an anatomical variant, as has already been postulated [2]. Probably, the individual anatomy of the abdominal wall can predispose an individual to this condition.

Of note, not many clinical surveys on this clinical problem have been performed in a pediatric population. Also, the results of the studies carried out among adults are inconsistent. Importantly, a majority of studies concerning dilatated linea alba were done in postpartum women, in whom this condition is probably acquired. Only some studies were performed on male patients and females with no history of pregnancy [26]. Fredon et al. correlated the anatomy of the anterior abdominal wall, which was assessed with computed tomography scans with the anthropometric data. They revealed that the rectus abdominis muscles and the linea alba structures differ between men and women [27]. In the study of Chiarello et al., the nulliparous women presented with the shortest IRD measured with ultrasound [28]. In the study of Dijvoh et al., the mean IRD value measured with ultrasound was 12 mm in nulliparous women and 14 mm in men, while the mean IRD value measured with calipers was 11 mm in nulliparous women and 13 mm in men [26]. Still, these differences between men and nulliparous women were neither statistically significant at rest (ultrasound measurement) nor in the head-lifted position (calipers measurement) [26]. Lee et al. did not find a difference in the IRD between male and female participants either, with the measurements conducted just above the umbilicus (U point) and halfway between the U point and the xiphoid (UX) point [29]. In the study of Gueroult et al., the linea alba width was independent of the gender of the participants [30].

In our study, the width of the linea alba was different in the measurement points. It was wider in the umbilicus and above the umbilicus than below the umbilicus. Similar results were obtained in the study conducted on adult women [31]. According to the results of Gueroult et al., the width of the linea alba increases with age and BMI [30]. By contrast, in our pediatric cohort, there was no association between BMI and increased IRD. Also, a more detailed analysis assessing the prevalence of DRA in children who were underweight, with normal weight, overweighted, or obese did not reveal significant correlations.

In our study, an IRD wider than 20 mm in any of the body positions was significantly more often found in children who had a higher waist/hip ratio (0.87 ± 0.03 vs. 0.90 ± 0.06; *p* = 0.009). In children, a higher IRD may be associated with a higher WHR, as a higher IRD usually indicates a relatively larger abdominal circumference in relation to the hip circumference. Clinically, DRA is more often found in patients with a rounded abdomen [3]. This phenomenon is due to DRA alone and not to obesity [3]. We acknowledge that there are several limitations to our research. A relatively small number of the participants seems to be the most important one. Secondly, there could be a selection bias resulting from the recruitment. Children were recruited through advertisements in social media and information disseminated at schools in our city. It could be assumed that the caregivers of those children who were previously diagnosed with DRA were more willing to take their child to be examined in this study.

On the other hand, there are a number of strong points in our survey. The ultrasonographic measurements were performed by the same person. The physical activity level was assessed with the use of the Physical Activity Questionnaire for Older Children (PAQ-C), which is considered to be a valid and reliable measure of general physical activity levels from childhood to adolescence [14,17,22,23]. This questionnaire has already been validated in the Polish population [14]. The anthropometric measurements were taken in accordance with the widely accepted recommendations. The OLAF calculator was used to assess the weight, height, and BMI, which is the tool dedicated to Polish children [18,19,20]. The additional advantage of this study was the use of the Tanita Body Composition Analyzer [21]. This analyzer can be used in healthy children aged 5–17 years old and also in healthy adults with different lifestyles, from active through moderately active to inactive [17]. The BIA (bioelectrical impedance analysis) augmented the anthropometric measurements, such as the weight and BMI, with the total body fat percent and weight, the total body water percent and weight, the total body muscle mass, and the fat-free mass (FFM). Body composition analysis has already been used in pediatric populations and has been proven to be a valuable diagnostic tool [24,25]. Moreover, we measured the width of the linea alba using ultrasonography. Such a measurement is objective and more accurate in comparison with the palpation, which requires a slight contraction of the rectus abdominis muscles and, therefore, should be performed in the head-lift position or during a curl-up but not in the resting position [26]. All the above-mentioned maneuvers are difficult to perform, especially in younger children.

We did not find an association between DRA and the level of physical activity. Several studies performed in adults, particularly in postpartum women, focused on the efficacy of dedicated physical exercises aimed at the reduction of IRD [32,33]. Still, the meta-analysis performed by Gluppe et al. demonstrated that scientific evidence that recommends specific exercise programs for the treatment of DRA in postpartum women is actually quite slim [34]. Although the medical literature claims that female patients with postpartum DRA can benefit from rehabilitation programs, further research is undoubtedly needed to find an appropriate and effective physiotherapeutic strategy [35].

The lack of association between physical activity and IRD in our study may be due to the small sample size, which limits the statistical power of analysis. Additionally, the measurements taken at a single time point could not capture the dynamic changes happening gradually during a longer period of time. In the long run, physical activity might influence changes in the linea alba width, which could be better assessed in a prospective study design. Another limitation of this study is the use of the PAQ-C questionnaire. It is a valid and reliable measure of general physical activity levels. However, it relies on self-reported data and may be subject to recall bias or inaccuracies in reporting the physical activity. In our cohort, we found a higher prevalence of DRA in children with a history of umbilical hernia diagnosed at birth. Umbilical hernias are relatively common in children. Such a hernia can be observed as a bulge in the area surrounding the newborn’s navel [25]. Most of these children are asymptomatic and do not require surgical treatment. A majority of these hernias resolve spontaneously in the first 5 years of life [36]. Ngom et al. reported 15 cases of hernias of the linea alba in a cohort of 450 children operated for umbilical hernias [37]. Wang et al. reported that umbilical hernia or DRA is relatively common in children presenting with Beckwith–Wiedemann syndrome [38]. Our results suggest that there is probably a relationship between congenital umbilical hernia (thus, the width of the umbilical ring in newborns) and the width of the IRD. However, regarding children, such an association has not yet been reported in the medical literature. Therefore, a selection bias regarding this variable cannot be ruled out. On the other hand, there are several reports of the coexistence of umbilical hernia and DRA in adult patients. According to Emanuelsson et al., some adult patients have umbilical hernias combined with DRA [39]. Similarly, Nishihara et al. noticed that in adult patients with DRA, the incidence of a coexisting umbilical hernia was high. Such patients were at a higher risk of recurrence of an umbilical hernia following surgical treatment [40]. In the study of Yuan et al., in adult female patients presenting with DRA, the most common type of hernia was an umbilical one [41].

From the embryological point of view, during the development of the fetus, the large umbilicus appears to stop myotomal cell migration in order to make a pair of myogenic cell pools of the abdominal band [42]. This can also be a potential explanation for the coexistence of DRA and an umbilical hernia.

Among the potential congenital factors influencing DRA is a hereditary genetic predisposition. There were not many genetic studies regarding diastasis recti in children. Digilio et al. reported vertical familial segregation of nonsyndromic anterior abdominal wall deficiency, characterized by diastasis recti and a weakness in the linea alba [43]. Among the proposed explanations was the autosomal dominant transmission of DRA [43]. These cases did not preset concomitant dysmorphic features or internal malformations [43].

Calvello et al. conducted a genetic study on Beckwith–Wiedemann syndrome (BWS), which is caused by various epigenetic and/or genetic alterations that dysregulate the imprinted genes on chromosome region 11p15.5. [44]. These patients often present with DRA. Calvello et al. performed a molecular analysis to reinforce the clinical diagnosis of BWS and to identify BWS patients with cancer susceptibility. They established a reliable molecular assay by pyrosequencing to quantitatively evaluate the methylation profiles of imprinting control region 1 (ICR1) and imprinting control region 2 (ICR2) and also explored the epigenotype–phenotype correlations [44]. They found a significant correlation (*p* < 0.001) between the mild percentage of imprinting control region 1 (ICR1) methylation (range: 55–59%) and BWS features, such as umbilical hernia and diastasis recti [44]. It should be underlined that patients whose ICR1 hypermethylation was mild (mean: 57%; range: 55–59%) had abdominal wall defects comprising an umbilical hernia or diastasis recti but not omphalocele [44].

Moreover, Wei et al. reported that the caldesmon gene (CALD1), located in the 7q33 region, is the locus associated with umbilical hernia development [45]. The CALD1 encodes caldesmon, a calmodulin- and actin-binding protein [45]. It plays an important role in the regulation of smooth muscle and non-muscle contractions [45]. The study of Wei et al. did not include diastasis recti. However, Wei et al. suggest common pathogenetic mechanisms for different subtypes of abdominal wall hernia (inguinal, femoral, umbilical, and ventral) [45]. The levels of genetic correlation of polygenic architecture in these pathologies were high [45]. However, this study did not include diastasis recti [45]. Of note, it should be underlined that a lack of a clear definition of DRA and the discrepancies between the IRD measurement procedures are still the biggest challenges when considering the research on this topic [9].

Finally, a significantly higher prevalence of DRA in boys should be discussed. In our study, we encountered more DRAs in boys. There are only a few studies comparing the linea alba width in boys and girls. We reported a similarly higher prevalence of DRA in boys in another study, which was performed in a different pediatric cohort [10]. Although, similarly to our previous study, a selection bias cannot be excluded, quite likely that gender plays a role in the pathogenesis of DRA. Since, in these children, DRAs were unlikely to be acquired, it may be hypothesized that genetic factors could be responsible. Potentially, the pathogenesis of DRA, similar to other pathologies of the connective tissue, may be linked to the FLNA gene. This gene is located on the X chromosome and encodes filamin A, which is a cytoplasmic protein that crosslinks actin filaments. Filamin A deficiency is associated with a number of pathologies, with some of them being related to the connective tissue and metabolism of collagen [46,47,48,49,50]. A higher prevalence of DRA in children with a higher body length at birth and a higher waist/hip ratio also suggests a genetic background linked to connective tissue. The FLNA gene is the only known gene located on the X chromosome that is involved in pathologies of the connective tissue. However, the genetic hypothesis regarding the FLNA gene needs stronger support and, to date, no scientific evidence is supporting such an association. Still, this possible genetic background should be validated by future studies.

Being aware of our study limitations, such as the small sample size and selection bias, we think that future studies should include more children in different age groups. Moreover, it would be valuable to include children younger than 7 years old. The choice of the age group in our study was related to the fact that the study aimed to correlate the anthropometric measurements assessed with BIA; the TANITA DC-430MA is dedicated to children older than 5 years [21]. The main novelty of this study was the assessment of the level of physical activity. This was thanks to the possibility of using the PAQ-C questionnaire designed for children older than 8 years old [14]. Moreover, children should be mature enough to participate in the study according to the established protocol of the ultrasonographic study. Children in Poland start primary school at the age of 7 years. Thus, the use of PAQ was possible. It was also easier to disseminate information about the planned survey at schools. We recruited children from all the primary schools in Opole. However, the area was limited to one region. Teachers can present different attitudes towards disseminating information about a study to parents. Also, as previously written, it could be assumed that the caregivers of children who were previously diagnosed with DRA were more willing to take their child to be examined. To avoid selection bias, recruitment should not be influenced by the preferences of the educators or parents. Future studies should explore this issue in pediatric cohorts, such as children from randomly chosen schools (all the children in particular age groups). Prospective studies aimed at observing the width of the linea alba over time in relation to physical activity would be of great value. The longitudinal studies, aimed at tracking changes in the IRD during a longer period of time, may help in designing preventive and therapeutic interventions. 

Our findings suggest that it is important for primary care physicians and pediatricians to pay attention to the structure of the anterior abdominal wall in children diagnosed with an umbilical hernia at birth. Perhaps parents should be instructed on how their child could perform exercises aimed at strengthening the muscles of the anterior abdominal wall in order to reduce the IRD. This issue requires attention and further research.

Educators should be aware of DRA. We did not examine the posture of the participants of the study. However, the literature indicates that these can be an association. Thus, physical education teachers and pediatricians should be alerted to the fact that children with postural defects could also present with DRA [13].

Clinical implications of the findings of our study can include a better understanding of the development of the linea alba and its relationship with the presented symptoms. The evaluation of IRD and early identification of DRA in children can implicate potential interventions in those at risk, for example, rehabilitation or physical activity. If the relationship between physical activity and IRD has been confirmed, it would be possible to propose an optimal and sufficient level of physical activity for children with DRA. Moreover, a clear definition of what can be considered a norm in pediatric populations can lead to a reduction in overdiagnosis and overtreatment. This would allow for a reduction in the costs associated with repeated ultrasound examinations and rehabilitation. Potential confounding factors in our study must be discussed. Several variables could influence the results but were not directly studied. For example, these factors comprise the socio-economic background of the families. We did not study the differences in income levels, parental education, or employment status. However, we admit that the level of education can affect the parents’ willingness to participate in the survey. The information was disseminated through schools, so access to the common education system was necessary to participate in this study. Children receiving homeschooling were not included in the study, yet in our country, such an education includes a very small number of children, particularly those in poor health conditions. Another confounding factor could be the physical and mental health of parents and children. Health issues (e.g., chronic illnesses and attention disorders) may significantly impact the possibility of participating in the study. Our study group was probably less representative than the broader population of children, primarily because of the above-described method of recruitment. It could be assumed that only children who receive better parental support participated in this study, which could also distort the study results. The advertisement and the questionnaires were written in Polish, so only Polish-speaking parents could participate in this study. Another potential confounding factor can be the residential area. Living in different parts of the region (e.g., in rural areas) also affected access to information about this study.

Our study is the first to evaluate the relationship between IRD and physical activity in children, making it a novel contribution to the field. Additionally, there is a lack of extensive research on this topic, highlighting the significance of this investigation in advancing our understanding of how these factors can be connected.

## 5. Conclusions

The results of our study indicate that there is no association between the presence of an increased IRD in children and the level of physical activity, obesity, body composition measured with a body analyzer, or the presence of abdominal symptoms. On the other hand, DRA was more prevalent in boys, children with a higher body length at birth, a higher waist/hip ratio, and also those with a history of congenital umbilical hernia.

## Figures and Tables

**Figure 2 pediatrrep-17-00025-f002:**
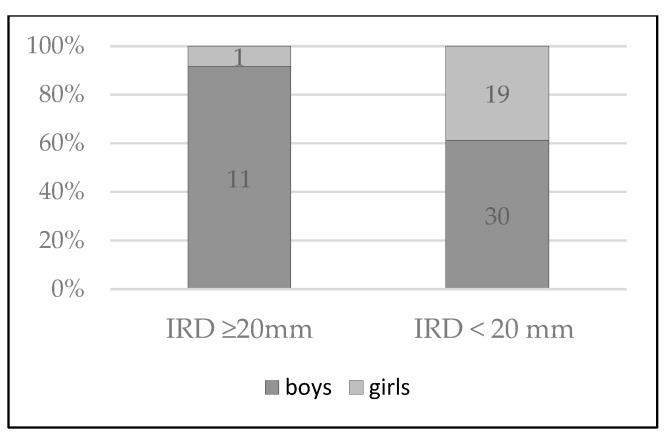
Enlarged interrectus distance (IRD) among boys and girls.

**Figure 3 pediatrrep-17-00025-f003:**
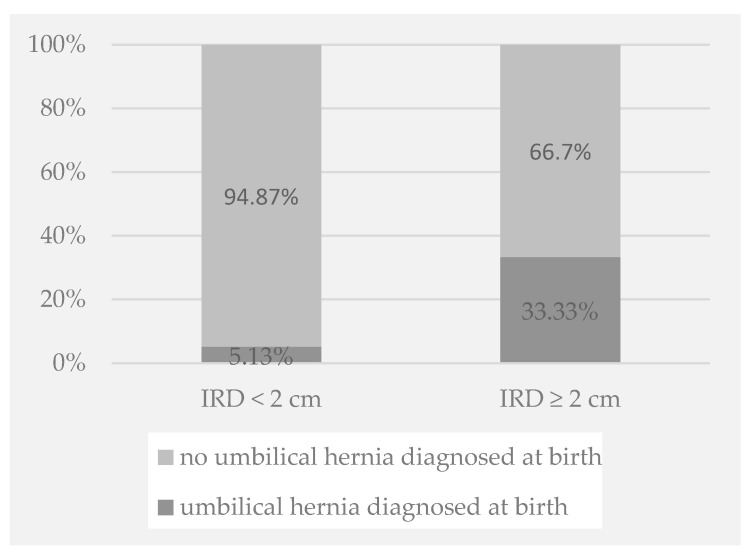
IRD and umbilical hernia diagnosed at birth.

**Figure 4 pediatrrep-17-00025-f004:**
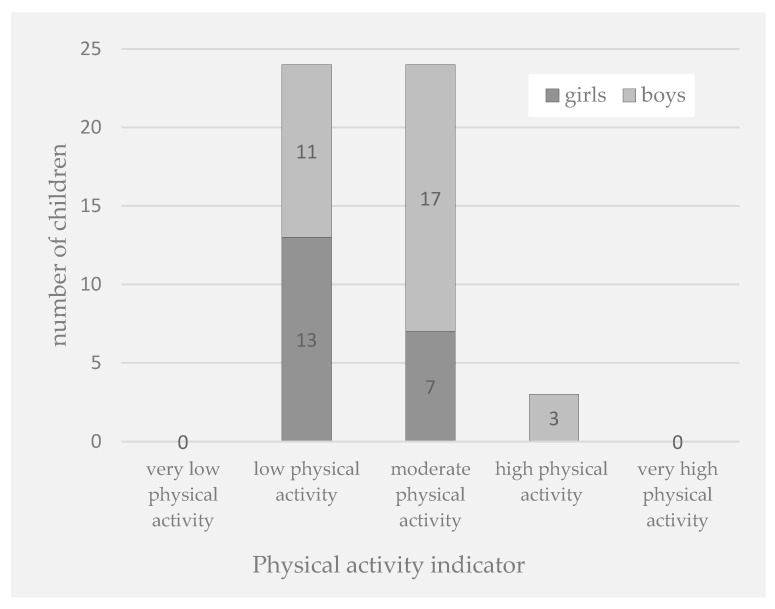
Physical activity level among boys and girls.

**Figure 5 pediatrrep-17-00025-f005:**
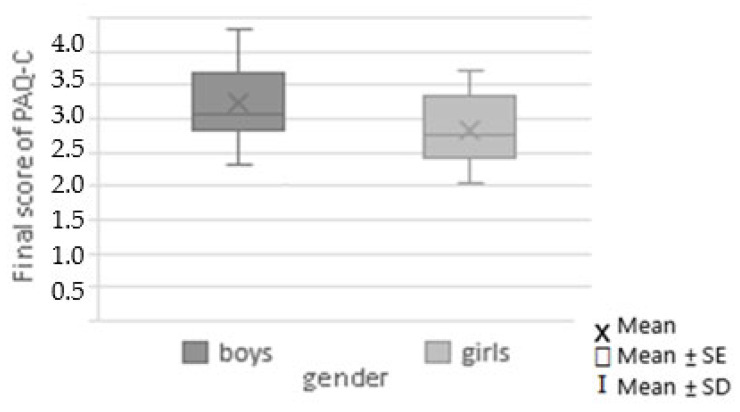
Final score of the PAQ-C according to the genders. The girls were less active than the boys (2.84 ± 0.53; median: 2.76; range: 2.05–3.72 vs. 3.23 ± 0.53; median: 3.06; range: 2.31- 4.34) (*p* = 0.019).

**Figure 6 pediatrrep-17-00025-f006:**
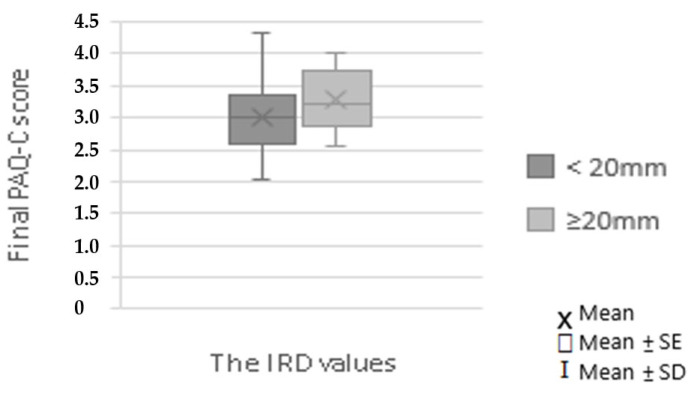
Physical activity level (assessed as the final PAQ-C score) among children with IRD ≥ 20 mm and IRD < 20 mm.

**Table 2 pediatrrep-17-00025-t002:** The ultrasonographic measurements of the interrectus distance.

Body Position; Measurement Point	Mean	Median	SD	Min	Max
Supine					
X	1.07	1.00	0.50	0.13	2.47
XU	1.41	1.32	0.53	0.46	2.53
U	1.48	1.37	0.62	0.56	3.20
PU	0.14	0.00	0.22	0.00	0.80
P	0.00	0.00	0.00	0.00	0.00
Sitting					
X	1.00	0.89	0.57	0.00	2.48
XU	1.42	1.32	0.59	0.00	3.25
U	1.53	1.49	0.58	0.67	3.20
PU	0.15	0.00	0.44	0.00	1.79
P	0.00	0.00	0.00	0.00	0.00
Standing					
X	1.02	0.92	0.53	0.19	2.73
XU	1.49	1.37	0.57	0.29	2.93
U	1.52	1.42	0.53	0.64	2.75
PU	0.08	0.00	0.18	0.00	0.81
P	0.00	0.00	0.00	0.00	0.00

**Table 3 pediatrrep-17-00025-t003:** Comparison of the anthropometric parameters in children with an interrectus distance wider than 20 mm in any of the body positions.

	IRD < 20 mm					IRD ≥ 20 mm					
	Mean	Median	SD	Min	Max	Mean	Median	SD	Min	Max	*p*
**Age (years)**	9.36	9.00	1.09	8.00	12.00	9.00	8.50	1.21	8.00	11.00	0.351
**Body mass** **at birth (g)**	3258.74	3400.00	555.78	1920.00	4360.00	3503.33	3455.00	313.32	3070.00	4300.00	0.067
**Body length** **at birth (cm)**	53.37	53.50	3.32	47.00	62.00	55.83	55.50	2.52	53.00	60.00	**0.023 ***
**PAQ-C ^1^**	3.02	3.02	0.57	2.05	4.34	3.28	3.22	0.49	2.56	4.03	0.155
**Body mass (kg)**	34.33	32.10	8.07	22.40	54.00	40.74	28.65	24.99	25.20	114.00	0.947
**Body mass** **(centile)**	48.28	48.00	28.14	0.10	96.00	56.74	51.50	33.75	2.00	99.90	0.242
**Height (cm)**	138.55	137.00	9.29	120.00	165.00	139.63	138.00	10.20	129.00	156.50	0.756
**Height** **(centile)**	45.46	43.00	29.43	1.00	97.00	54.67	56.50	28.90	16.00	96.00	0.256
**BMI ^2^**	17.68	17.29	2.53	13.46	25.77	20.06	17.19	9.09	13.17	46.84	0.938
**BMI ^2^** **(centile)**	51.90	53.00	26.79	0.10	98.00	56.67	62.50	36.21	0.10	99.90	0.323
**Abdominal circumference (cm)**	64.54	63.00	7.86	52.00	83.00	64.50	63.00	7.67	54.00	79.00	0.453
**Waist circumference (cm)**	59.96	58.00	6.00	50.00	74.00	62.17	60.50	7.99	51.00	77.00	0.260
**Hip circumference (cm)**	69.23	68.00	7.41	59.00	88.00	68.92	68.50	8.39	57.00	82.00	0.593
**WHR ^3^**	0.87	0.87	0.03	0.80	0.93	0.90	0.90	0.06	0.81	1.03	**0.009 ***
**Fat% ^4^**	19.65	18.00	6.48	10.90	36.70	17.52	17.65	5.98	8.90	28.50	0.314
**Fat mass ^5^**	7.42	6.00	4.47	2.50	22.90	6.64	5.65	3.73	2.50	14.30	0.564
**FFM ^6^**	26.87	26.20	6.13	6.90	43.80	29.21	27.40	6.63	21.60	41.10	0.386
**Muscle mass**	25.86	24.80	4.91	18.70	41.50	27.64	25.95	6.33	20.40	39.00	0.457
**tbw ^7^**	20.13	19.50	3.79	14.60	32.10	21.38	20.05	4.86	15.80	30.10	0.477
**tbw% ^7^**	57.92	59.90	8.13	17.10	65.30	60.39	60.40	4.32	52.50	66.50	0.380

* The statistically significant results are highlighted in bold and marked with an asterisk: ^1^ PAQ-C (The Physical Activity Questionnaire for Older Children) activity summary score; ^2^ BMI—body mass index (kg/m2); ^3^ WHR—waist-to-hip ratio; ^4^ Fat%—body fat %; ^5^ Body fat mass (fat mass); ^6^ FFM—fat-free mass; ^7^ TBW—total body water.

**Table 4 pediatrrep-17-00025-t004:** The body composition and physical activity level. Statistically significant results are highlighted in bold.

	PAQ-C ^1^ < 3					PAQ-C ^1^ ≥ 3					
	Mean	Median	SD	Min	Max	Mean	Median	SD	Min	Max	*p*
Fat % ^2^	20.86	20.55	6.71	10.90	36.70	17.63	17.20	5.51	8.90	30.30	0.115
Fat mass ^3^	8.73	7.75	4.91	2.70	22.90	5.91	4.80	2.98	2.50	14.90	**0.025 ***
Ffm ^4^	28.75	29.10	6.96	6.90	41.10	26.23	25.60	5.16	19.90	43.80	**0.041 ***
Muscle mass	27.91	27.55	5.09	20.70	39.00	24.84	24.20	4.90	18.70	41.50	**0.022 ***
tbw ^5^	21.76	21.70	3.86	16.00	30.10	19.24	18.70	3.77	14.60	32.10	**0.015 ***
tbw% ^5^	56.49	58.05	9.45	17.10	65.30	60.29	60.50	4.05	50.90	66.50	0.101
BMI ^6^	20.13	17.95	8.05	15.00	56.40	17.08	16.30	2.27	13.50	23.20	**0.046 ***
Muscle mass %	75.33	76.27	6.07	59.41	84.49	79.04	78.31	7.93	65.99	109.82	0.132

* Statistically significant results are highlighted in bold and marked with an asterisk. ^1^ PAQ-C (The Physical Activity Questionnaire for Older Children) activity summary score; ^2^ Fat%—body fat %; ^3^ body fat mass (Fat mass); ^4^ FFM—fat-free mass; ^5^ TBW—total body water; ^6^ BMI—body mass index (kg/m^2^).

**Table 5 pediatrrep-17-00025-t005:** The IRD measurements and genders. Statistically significant results are highlighted in bold.

Body Position; Measurement Point	Gender	Mean	Median	SD	Min	Max	*p*
Supine XU	boys	1.54	1.41	0.52	0.69	2.53	**0.013 ***
	girls	1.21	1.24	0.47	0.46	1.93	
Supine U	boys	1.75	1.69	0.61	0.75	3.20	**<0.001 ***
	girls	1.07	1.06	0.35	0.56	1.84	
Sitting XU	boys	1.58	1.43	0.65	0.00	3.25	**0.003 ***
	girls	1.18	1.23	0.36	0.48	1.77	
Sitting U	boys	1.75	1.75	0.55	0.72	3.20	**<0.001 ***
	girls	1.20	1.11	0.46	0.67	2.70	
Standing XU	boys	1.66	1.69	0.58	0.70	2.93	**0.002 ***
	girls	1.23	1.28	0.43	0.29	1.99	
Standing U	boys	1.72	1.58	0.50	0.97	2.75	**<0.001 ***
	girls	1.21	1.11	0.41	0.64	2.20	

* Statistically significant results are highlighted in bold and marked with an asterisk.

## Data Availability

The data presented in this study are available on request from the corresponding author due to privacy, legal, and ethical reasons.

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
