# Peer review of "The Linea Alba Width, Children’s Physical Activity, and Chosen Anthropometric Measurements: The Results of the Cross-Sectional Study"

_pediatrrep, 2025, doi:10.3390/pediatric17010025_

Round 1

Reviewer 1 Report

Comments and Suggestions for Authors

Thank you for considering for considering Pediatr. Rep. for submission.

Please, consider the peer review comments in the attachment. 

Best regards

Comments on the Quality of English Language

Please, improve quality of English. Some sentences are overly complex or repetitive. Simplify language where possible to improve readability.

Author Response

Dear Reviewer,

Thank you very much for your thoughtful and thorough review of our article titled “The linea alba width and children’s physical activity and chosen anthropometric measurements ‒ the results of the cross-sectional study" We greatly appreciate the time and effort you put into evaluating our work, as well as your valuable feedback.

Your comments were taken them into consideration when revising the article. Regarding your suggestions we added more paragraphs. I believe these adjustments will strengthen the overall argument and clarity of the manuscript.

I have attached a file containing detailed responses to the comments and suggestions you provided on our manuscript. We have carefully addressed each point raised, and we believe the revisions have strengthened the work.

Once again, thank you for your constructive critique. I hope the revised version of the article will meet the journal's standards.

Best regards,

Author

Specific comments:

Introduction:

 The introduction provides a general overview of DRA but lacks specificity in connecting the pediatric focus to existing literature. The authors should:

▪ Clearly define DRA and IRD early in the introduction and contextualize their relevance in children.

The following paragraphs were added:

It is described as a separation of the rectus abdominis muscles along the linea alba that can result in a midline abdominal bulge [3]. It must be underlined that this condition is not a hernia and there is no risk of incarceration [4]. It is crucial to distinguish whether the fascia is intact or disrupted because this is a difference between DRA and hernia [3].

Diagnostic techniques for diagnosing DRA include physical examination and imaging studies [3]. The recommended way is to examine the patients in the supine position and ask them to engage their core (a half sit-up position or with a leg raise) [3]. In most cases of DRA in patients with normal body habitus, a focused physical examination is sufficient to confirm the diagnosis of DRA [3-6]. It can be more difficult in obese patients [3]. Among the tools used in the diagnosis, the digital calipers and diastomameters were described [7]

The imaging studies used in diagnosis include the use of ultrasound, computed tomography (CT), and magnetic resonance imaging (MRI) [3]. Ultrasound is considered the cheapest and the most accessible imaging modality worldwide [3]. When considering measurement in children, it is worth emphasizing that this method does not require anesthesia and does not expose children to ionizing radiation. According to Petronila et al. palpation techniques, digital calipers, and diastometers are reliable tools but cannot be substitutes for ultrasound imaging for the clinical measurement of diastasis rectus abdominis [7].

In imaging studies, the severity of DRA is determined by measuring the interrectus distance (IRD) [3-6]. The IRD is defined as the width of the linea alba between the connective tissue sheaths surrounding the paired rectus abdominis muscles [8].

To visualize the rectus abdominis separation in the ultrasound a high-frequency linear array probe should be used (muscular preset)[6]. During measurement, the patient should be resting [6]. This means lying in the supine position with the abdominal muscles fully relaxed [6]. The probe plane should be perpendicular to the long axis of the abdomen [6]. In most studies, the IRD investigation was located at the umbilicus and 2 or 3 cm above and below the umbilicus [6; 9].

It must be underlined that in adults and children, there are no generally accepted criteria for diagnosis of diastasis of the rectus abdominis muscle [4]. The need for DRA screening method standardization and IRD measurement protocol standardization are widely discussed [9]. Opala et al. conducted a scoping review according to PRISMA-ScR guidelines and noticed the discrepancies between the IRD measurement procedures which enables performing between-study comparisons [9].

There are many classifications to assess the severity of DRA in adults, for example, the Rath classification, and the Nahas classification [3]. So far, there are no classifications for children. In adult patients, IRD 2 cm or greater is considered clinically significant [3]. In the literature, the cut-off points of IRD value in children such as 15 mm and 20 mm were considered [10]. The choice of 20 mm as the cut-off point seems more appropriate because Cohen’s kappa coefficient for the diagnosis of DR using the clinical definition and ultrasonographic criterion (20 mm) was higher than for the ultrasonographic criterion (15 mm), indicating higher agreement between these two modes of diagnosing [10].

▪ Justify the selection of the 20 mm threshold for IRD more robustly, referencing previous studies and clinical significance.

It was added in Materials and Methods”:

For the purpose of this survey, the enlarged IRD was defined as wider than 20 mm at any of the above-described body positions and parts of the linea alba.

The choice of this cut-off point was driven by the results of previous study, which demonstrated that Cohen’s kappa coefficient for the diagnosis of DR using the clinical definition and ultrasonographic criterion (20 mm) was higher than for the ultrasonographic criterion (15 mm) (0.49 vs. 0.32), indicating higher agreement between these two modes of diagnosing [10]. A higher kappa suggests that the assessment process is more reliable and consistent, reducing potential for errors or discrepancies in the data.

Also, elaborate on the gaps in current pediatric research on DRA and how this study aims to address them.

The following paragraphs were added:

Although DRA is considered to be an acquired condition [1], it can also be found in children. In comparison with the adults, data on DRA in pediatric population are even more limited. It is known that DRA is more prevalent in pediatric patients presenting with genetic disorders. Still, an enlarged distance between two rectus abdominis muscles can also be found among otherwise healthy children.

DRA seems to be a quite common finding in children, especially among boys [10]. There is little information in the literature regarding the clinical significance of the presence of DRA in children. It is described that only a few healthy children with DRA reported abdominal pain after physical exertion, constipation, fecal and urinary incontinence but statistically sound conclusions could not be drawn [10]. It was demonstrated that the width of the linea alba in children can be related to the spine's shape [13]. Zmyślna et al. found that children with DRA present with a posterior shift of the body axis [13]. Consequently, DRA can be considered a sign of poor coordination of the abdominal muscles, which are important stabilizers of the spine [13]. In current pediatric research on DRA, there are no studies comparing the body composition of children with and without DRA and their physical activity. This knowledge gap may lead to a lack of consistent consensus among pediatricians, and physiotherapists, confusion among caregivers of children, as well as overtreatment and rehabilitation of healthy children. This study was aimed at assessment of the relationship between the presence of DRA in children and other clinical parameters, such as anthropometric measurements, physical activity level, family history of musculoskeletal diseases, and other clinical variables potentially attributable to DRA.

 Material and Methods:

 ▪ Recruitment Process: Provide a more detailed explanation of the recruitment strategy and how bias was minimized. For example, clarify whether advertisements targeted specific demographics.

The following paragraphs were added:

The recruitment process included sending the information about the study to the secretariats of 23 public primary schools in Opole and disseminating the information on a Facebook page dedicated to mothers of children from Opole (Mamy z Opola group). The caregivers interested in participating in the study used the study's website to answer questions about the exclusion criteria and choose the date and hour of the examination. To avoid bias clear inclusion and exclusion criteria were applied before the participant recruitment, the information was send to all the primary schools in the region, the study’s purpose and procedures were explained to participants and their guardians to encourage informed and voluntary participation.

Ultrasonographic Measurements: Include information about inter-rater reliability or how consistency was ensured, as measurements were conducted by a single operator.

The following paragraphs were added:

To maintain consistency the same protocol and techniques were followed during each examination. Firstly, the patients were examined on the same examination table in the supine position, the measurement sites were marked with the body marker. During measurement, the probe plane was perpendicular to the long axis of the abdomen, and image acquisition was performed at the end of normal expiration. The use of a mean IRD value of three images per measurement site as measurement outcome. Secondly, the examination was performed in the sitting and standing positions in the site marked previously. The physician conducting the ultrasounds was trained to minimize variability in the way the scans were performed, passed the exam of the Polish Ultrasound Society, and holds certification to perform pediatric ultrasound examinations.

Physical Activity Assessment: Explain why the PAQ-C was chosen over alternative methods and discuss its limitations in capturing physical activity intensity or type.

The following paragraphs were added in Material and Methods:

We also assessed the level of physical activity of each child, using the the PAQ-C (The Physical Activity Questionnaire for Older Children). The items of this questionnaire were scored on the 5-point scale. Then the final PAQ-C activity summary score was calculated. A score of 1 indicated a low physical activity, whereas a score of 5 indicated a high physical activity [14, 17].

The Physical Activity Questionnaire for Older Children was already validated for the Polish population and showed a high reliability and internal consistency in measuring general physical activity levels in youth [14-17; 22-25].

The following paragraphs were added in Discussion:

On the other hand, there is a number of strong points in our survey. The ultrasonographic measurements were performed by the same person. The physical activity level was assessed with the use of the Physical Activity Questionnaire for Older Children (PAQ-C), which is considered to be a valid and reliable measure of the general physical activity level from the childhood to the adolescence [-14, 17, 22, 23]. This questionnaire has been already validated in the Polish population

Results

The results section is detailed but could benefit from better organization and presentation:

▪ Highlight key findings more prominently, particularly the statistically significant associations with IRD (e.g., gender, birth length, and waist-hip ratio).

It was improved the tables * were added.

The Discussion starts:

In this study we found that an increased IRD was more prevalent in boys, in children with a higher body length at birth, and presenting with a higher waist-hip ratio.

  • Ensure that all tables and figures are fully self-explanatory. For example, provide more descriptive captions for Tables 1-4 and Figures 1-6 to guide interpretation. Consider improving the quality of Figure 1 (ultrasonographic image) for better clarity. ▪ Discuss non-significant findings, such as the lack of association between physical activity and IRD, to contextualize these results and suggest possible reasons (e.g., limited sample size).

The tables were improved adding legends and explanation of abbreviations.

The following paragraphs were added in Discussion

We acknowledge that there are several limitations of our research. A relatively small number of the participants seems to be the most important one. Secondly, there could be a selection bias resulting from the recruitment. Children were recruited through the advertisements in social media and information disseminated at all the primary schools in Opole. It could be assumed that the caregivers of those children who were previously diagnosed with DRA, were more willingly to take their child to be examined in this study.

The lack of association between physical activity and IRD in our study may be due to the small sample size, which limits the statistical power of the analysis. Additionally, the measurements taken at a single time point may not capture dynamic changes over time. Physical activity might influence these changes in linea alba width over time, which could be better assessed in a prospective study design. Another limitation of the study is the use of the PAQ-C questionnaire. It is a valid and reliable measure of the general physical activity level. However, it relies on self-reported data and may be subject to recall bias or inaccuracies in reporting physical activity levels.

Discussion

The discussion effectively connects findings to existing literature but requires deeper exploration:

▪ Expand on the potential congenital factors influencing DRA, referencing genetic studies where possible.

The following paragraphs were added in Discussion:

From the embryological point of view, during the development of the fetus, the large umbilicus appears to stop myotomal cell migration in order to make a pair of myogenic cell pools or the abdominal band [42]. This can also be a potential explanation for the co-existence of DRA and umbilical hernia.

Among the potential congenital factors influencing DRA is hereditary genetic predisposition. There were not many genetic studies regarding diastasis recti in children. Digilio et al. reported vertical familial segregation of nonsyndromic anterior abdominal wall deficiency characterized by diastasis recti and weakness of the linea alba [43]. Among the proposed explanations was the autosomal dominant transmission of DRA [43].  The cases did not preset concomitant dysmorphic features or internal malformations [43].

Calvello et al. conducted genetic study on Beckwith-Wiedemann syndrome (BWS) caused by various epigenetic and/or genetic alterations that dysregulate the imprinted genes on chromosome region 11p15.5. [44]. These patients often present DRA. Calvello et al. performed molecular analysis to reinforce the clinical diagnosis of BWS and to identify BWS patients with cancer susceptibility. They established a reliable molecular assay by pyrosequencing to quantitatively evaluate the methylation profiles of imprinting control region 1 (ICR1) and imprinting control region 2 (ICR2) and explored epigenotype-phenotype correlations [44]. They found a significant correlation (p<0.001) between the mild percentage of imprinting control region 1 (ICR1) methylation (range: 55-59%) and BWS features such as umbilical hernia and diastasis recti [44]. It must be underlined that patients which ICR1 hypermethylation was mild (mean: 57%; range: 55–59%) had abdominal wall defects comprising umbilical hernia or diastasis recti, but not omphalocele [44].

Moreover, Wei et al. reported that the caldesmon gene (CALD1), located in the 7q33 region, is a locus associated with umbilical hernia development [45]. CALD1 encodes caldesmon, a calmodulin- and actin-binding protein [45]. It plays an important role in the regulation of smooth muscle and non-muscle contraction [45]. The study of WEI et al. did not include diastasis recti. Wei et al. suggest common pathogenetic mechanisms for different subtypes of abdominal wall hernia (inguinal, femoral, umbilical, ventral) [45]. The levels of genetic correlation of polygenic architecture in these pathologies were high [45]. However, the study did not include the diastasis recti [45]. It must be underlined, that lack of a clear definition of diastasis recti and the discrepancies between the IRD measurement procedures is the biggest challenge when considering the research on this topic [9].

Finally, significantly higher prevalence of DRA in boys should be discussed. In our study we encountered more DRAs in boys. There are only a few studies comparing the linea alba width in boys and girls. We reported a similarly higher prevalence of DRA in boys in another study, which was performed at a different pediatric cohort [10]. Although, similarly to our previous study, a selection bias cannot be excluded, quite likely the gender plays a role in the pathogenesis of DRA. Since in these children DRAs were unlikely to be acquired, it may be hypothesized that genetic factors could be responsible. Potentially, the pathogenesis of DRA, similarly to other pathologies of the connective tissue, may be linked to the FLNA gene. This gene is located on the X chromosome and is encoding the filamin A, which is a cytoplasmic protein that crosslinks the actin filaments. Filamin A deficiency is associated with a number of pathologies, some of them related to the connective tissue and metabolism of the collagen [46, 47, 48, 49 50]. A higher prevalence of DRA in children with a higher body length at birth and a higher waist-hip ratio also suggests a genetic background linked to the connective tissue. The FLNA gene is the only known gene located on the X chromosome that is involved in pathologies of the connective tissue. However, the genetic hypothesis regarding FLNA gene needs stronger support. Still, this possible genetic background should be validated by future studies.

  • ▪ Discuss the clinical implications of the findings, especially for early identification and potential interventions for children at risk of DRA.
  • ▪ Provide a more critical analysis of study limitations, including the potential for selection bias and the relatively small sample size.
  • ▪ Suggest future research directions, such as longitudinal studies to track changes in IRD over time or genetic analyses to explore hereditary factors.

The following paragraphs were added in Discussion

Being aware of our study limitations such as small sample size and selection bias, we think that future studies should include more children also in different age groups. Moreover, it would be valuable to include children younger than 7 years old in the study group. The choice of the age group in our study was related to the fact that the study aimed to correlate the anthropometric measurements assessed with BIA. TANITA DC-430MA is dedicated to children over 5 years old [21]. The main novelty of the study was the assessment of the level of physical activity. This was thanks to the possibility of using the PAQ-C questionnaire designed for children over 8 years old [14]. Moreover, the age of the children must be mature enough to participate in the study according to the established protocol of the ultrasonographic study. Children in Poland start attending primary school at 7 years old. Thus, the use of PAQ was possible. It was also easier to disseminate the information at schools. We recruited children from all the primary schools in Opole. However, the area was limited to one region. The teachers can present different attitudes towards disseminating the information about the study to the parents. Also, as previously written, it could be assumed that the caregivers of children who were previously diagnosed with DRA, were more willing to take their child to be examined in this study. To avoid selection bias the recruitment should not be influenced by the preference of the educators or parents. Future studies should explore this issue in pediatric cohorts, such as children from randomly chosen schools (all the children in particular age groups).Prospective studies aimed at observing the width of the linea alba over time in relation to physical activity would be of great value. The longitudinal studies aimed to track changes in IRD over time may help design preventive and therapeutic interventions.  

Our findings suggest that it is important for primary care physicians and pediatricians to pay attention to the structure of the anterior abdominal wall in children diagnosed with an umbilical hernia at birth. Maybe parents should be instructed to have their child perform exercises aimed at strengthening the muscles of the anterior abdominal wall, to reduce the IRD. This issue requires attention and further research.

Educators should be aware of DRA. We did not examine the posture of the participants of the study. However, the literature indicates that it can be linked. Thus, physical education teachers and pediatricians should be alerted to the fact that children with postural defects should be suspected of having DRA [13].

The clinical implications of the findings of our study may include a better understanding of the development of linea alba and its relationship with presented symptoms. The evaluation of IRD and the early identification of DRA in children can implicate potential interventions for children at risk of DRA, for example regarding rehabilitation or physical activity. If the relationship between physical activity and IRD has been confirmed, it would be possible to propose an optimal and sufficient level of physical activity for children with DRA. Moreover, a clear definition of what can be considered a norm in the pediatric population can lead to a reduction in overdiagnosis and overtreatment. This would allow for a reduction in costs associated with repeated ultrasound examinations and rehabilitation expenses. Potential confounding factors in our study must be discussed. The various variables can influence the results but were not directly studied. Some examples of these factors are the socio-economic background of the families. We did not study differences in income levels, parental education, or employment status. However, we admit that the level of education can affect the parents' will to participate in the project. The information was disseminated through schools, so access to educational resources was crucial to being a participant in the study. Children receiving homeschooling were not included in the study. Another confounding factor can be the physical and mental health of parents and children. Health issues (e.g., chronic illnesses, and attention disorders) may significantly impact the possibility of participating in the study. The study group can be less representative of the broader population of children because parents needed to declare their willingness to participate in the study and fill out the questionnaire. It can be assumed that only children who receive greater parental support participated in the study, which could affect study results. The advertisement and the questionnaires were written in Polish, so only Polish-speaking parents could participate in the study. Another potential confounding factor can be the residential area. Living in different parts of the region (e.g., in rural areas) also affected access to the information about the study.

Our study is the first to evaluate the relationship between IRD and physical activity in children, making it a novel contribution to the field. Additionally, there is a lack of extensive research on this topic, highlighting the significance of this investigation in advancing our understanding of how these factors can be connected.

Reviewer 2 Report

Comments and Suggestions for Authors

Dear Editor,

Thank you for inviting me to review manuscript ID pediatrrep-3413259 entitled "The linea alba width and children's physical activity and chosen anthropometric measurements ‒ the results of the cross-sectional study". This cross-sectional study investigated potential associations between diastasis recti abdominis (DRA) and various clinical parameters in children aged 8-12 years. The main findings indicate that DRA was more prevalent in boys, children with higher birth length and waist-hip ratio, and those with a history of congenital umbilical hernia. Notably, no associations were found between DRA and physical activity levels, body composition, or other clinical parameters.

This manuscript addresses an important knowledge gap regarding DRA in pediatric populations and provides novel insights into potential congenital/anatomical factors. While the study is generally well-conducted, several aspects require attention before publication.

General Comments:

  1. Statistical Analysis:
  • The sample size (n=51) appears relatively small for meaningful statistical comparisons
  • The statistical approach needs more detailed justification
  • Consider performing power calculations to support the sample size
  • Multiple testing corrections should be addressed
  1. Study Design:
  • The cross-sectional nature limits causal inference
  • Selection bias needs more thorough discussion
  • Consider addressing potential confounding factors
  1. Results Presentation:
  • Some figures could be improved for clarity
  • Tables contain excessive decimal places
  • Statistical significance reporting needs standardization
  1. Discussion:
  • The genetic hypothesis regarding FLNA gene needs stronger support
  • Clinical implications require more detailed discussion
  • Study limitations should be expanded

Specific Comments:

Introduction:

  • Provide more context about DRA prevalence in children
  • Clarify the rationale for choosing the specific age range (8-12 years)
  • Include more recent references about DRA assessment methods

Methods:

  • Justify the 20mm cutoff value for defining DRA
  • Detail the reliability assessment of ultrasonographic measurements
  • Explain the selection of measurement points more thoroughly
  • Provide more details about the PAQ-C questionnaire validation

Results:

  • Present confidence intervals for main findings
  • Include effect sizes where appropriate
  • Clarify the handling of missing data
  • Provide a more detailed analysis of gender differences

Discussion:

  • Expand on the clinical relevance of findings
  • Compare results more thoroughly with existing literature
  • Discuss implications for pediatric assessment and treatment
  • Address the potential impact of growth and development

Conclusions:

  • Should better reflect study limitations
  • Need more specific recommendations for clinical practice
  • Include suggestions for future research
  •  
  •  

In conclusion, while this manuscript presents interesting findings about DRA in children, substantial revisions are needed to improve its scientific rigor and clinical relevance. The authors should particularly focus on strengthening their statistical analysis, addressing potential biases, and expanding the discussion of clinical implications.

Author Response

Dear Reviewer,

Thank you very much for your thoughtful and thorough review of our article titled The linea alba width and children’s physical activity and cho-sen anthropometric measurements ‒ the results of the cross-sectional study" We greatly appreciate the time and effort you put into evaluating my work, as well as your valuable feedback.

Your comments were taken them into consideration when revising the article.

Regarding your suggestions we added more paragraphs.

I believe these adjustments will strengthen the overall argument and clarity of the manuscript.

I am attaching a file with detailed responses to the comments.

Once again, thank you for your constructive critique. I look forward to submitting the revised version of the article and hope it will meet the journal's standards.

Best regards,

Round 2

Reviewer 2 Report

Comments and Suggestions for Authors

Dear Editor,

I carefully reviewed the second version of the manuscript entitled "The linea alba width and children's physical activity and chosen anthropometric measurements  the results of the cross-sectional study" (ID: pediatrrep-3413259). After reviewing the revised manuscript, I find the authors have successfully addressed the major concerns raised in the initial review. 

The improvements are substantial and thorough the manuscript. The statistical methodology now includes appropriate power calculations and sample size considerations. The ultrasonographic measurement protocols are detailed and include reliability assessments. The discussion section has been significantly enhanced with comprehensive coverage of genetic factors and embryological background. The handling of missing data is well documented, and study limitations are thoroughly addressed. Clinical implications have been strengthened, and the figures show improved statistical reporting.

While some minor elements could be further refined (statistical reporting consistency, table formatting, and clinical implementation details), these do not diminish the manuscript's scientific contribution or validity. The authors have demonstrated a rigorous approach to addressing previous concerns and have produced a valuable contribution to pediatric DRA research. For all reasons mentioned above, my decision is accepted in present form.

The manuscript in its current form makes a significant contribution to the understanding of DRA in pediatric populations. The authors have successfully addressed the major concerns from the initial review, and the remaining minor issues do not warrant further revision. The study provides valuable insights that will benefit both researchers and clinicians in this field.

Kindly regards,

The reviewer